# Continuous Meta-Learning without Tasks

**James Harrison, Apoorva Sharma, Chelsea Finn, Marco Pavone**
Stanford University, Stanford, CA
{jharrison, apoorva, cbfinn, pavone}@stanford.edu

## Abstract

Meta-learning is a promising strategy for learning to efficiently learn using data gathered from a distribution of tasks. However, the meta-learning literature thus far has focused on the *task segmented* setting, where at train-time, offline data is assumed to be split according to the underlying task, and at test-time, the algorithms are optimized to learn in a single task. In this work, we enable the application of generic meta-learning algorithms to settings where this task segmentation is unavailable, such as continual online learning with unsegmented time series data. We present meta-learning via online changepoint analysis (MOCA), an approach which augments a meta-learning algorithm with a differentiable Bayesian changepoint detection scheme. The framework allows both training and testing directly on time series data without segmenting it into discrete tasks. We demonstrate the utility of this approach on three nonlinear meta-regression benchmarks as well as two meta-image-classification benchmarks.

## 1 Introduction

Meta-learning methods have recently shown promise as an effective strategy for enabling efficient few-shot learning in complex domains from image classification to nonlinear regression [10, 40]. These methods leverage an offline meta-learning phase, in which data from a collection of learning tasks is used to learn priors and update rules for more efficient learning on new related tasks.

Meta-learning algorithms have thus far solely focused on settings with *task segmentation*, where the learning agent knows when the latent task changes. At meta-train time, these algorithms assume access to a meta-dataset of datasets from individual tasks, and at meta-test time, the learner is evaluated on a single task. However, there are many applications where task segmentation is unavailable, which have been under-addressed in the meta-learning literature. For example, environmental factors may change during a robot's deployment, and these changes may not be directly observed. Furthermore, crafting a meta-dataset from an existing stream of experience may require a difficult or expensive process of detecting switches in the task.

In this work, we aim to enable meta-learning in task-unsegmented settings, operating directly on time series data in which the latent task undergoes discrete, unobserved switches, rather than requiring a pre-segmented meta-dataset. Equivalently, from the perspective of online learning, we wish to optimize an online learning algorithm using past data sequences to perform well in a sequential prediction setting wherein the underlying data generating process (i.e. the task) may vary with time.

**Contributions.** Our primary contribution is an algorithmic framework for task unsegmented meta-learning which we refer to as meta-learning via online changepoint analysis (MOCA). MOCA wraps arbitrary meta-learning algorithms in a differentiable Bayesian changepoint estimation scheme, enabling their application to problems that require continual learning on time series data. By backpropagating through the changepoint estimation framework, MOCA learns both a rapidly adaptive underlying predictive model (the meta-learning model), as well as an effective changepoint detection algorithm, optimized to work together. MOCA is a generic framework which works with

many existing meta-learning algorithms. We demonstrate MOCA on both regression and classification settings with unobserved task switches.

## 2 Problem Statement

Our goal is to enable meta-learning in the general setting of sequential prediction, in which we observe a sequence of inputs $x_t$ and their corresponding labels $y_t$. In this setting, the learning agent makes probabilistic predictions over the labels, leveraging past observations: $p_{\boldsymbol{\theta}}(\hat{\boldsymbol{y}}_t \mid \boldsymbol{x}_{1:t}, \boldsymbol{y}_{1:t-1})$, where $\boldsymbol{\theta}$ are the parameters of the learning agent. We assume the data are drawn from an underlying generative model; thus, given a training sequence from this model $\mathcal{D}_{\text{train}} = (\boldsymbol{x}_{1:N}, \boldsymbol{y}_{1:N})$, we can optimize $\boldsymbol{\theta}$ to perform well on another sample sequence from the same model at test time.

We assume data is drawn according to a latent (unobserved) *task* $\mathcal{T}_t$, that is $\boldsymbol{x}_t, \boldsymbol{y}_t \sim p(\boldsymbol{x}, \boldsymbol{y} \mid \mathcal{T}_t)$. Further, we assume that every so often, the task switches to a new task sampled from some distribution $p(\mathcal{T})$. At each timestep, the task changes with probability $\lambda$, which we refer to as the hazard rate. We evaluate the learning algorithm in terms of a log likelihood, leading to the following objective:

$$\min_{\boldsymbol{\theta}} \quad \mathbb{E}\left[\sum_{t=1}^{\infty} - \log p_{\boldsymbol{\theta}}(\boldsymbol{y}_t \mid \boldsymbol{x}_{1:t}, \boldsymbol{y}_{1:t-1})\right] \tag{1}$$
$$\text{subj. to} \quad \boldsymbol{x}_t, \boldsymbol{y}_t \sim p(\boldsymbol{x}, \boldsymbol{y} \mid \mathcal{T}_t),$$
$$\mathcal{T}_t = \begin{cases} \mathcal{T}_{t-1} & \text{w.p. } 1 - \lambda \\ \mathcal{T}_{t,\text{new}} & \text{w.p. } \lambda \end{cases}, \quad \mathcal{T}_1 \sim p(\mathcal{T}), \quad \mathcal{T}_{t,\text{new}} \sim p(\mathcal{T})$$

Given $\mathcal{D}_{\text{train}}$, we can approximate this expectation and thus learn $\boldsymbol{\theta}$ at train time.

Note that just as in standard meta-learning, we leverage data drawn from a diverse collection of tasks in order to optimize a learning agent to do well on new tasks at test time. However, there are three key differences from standard meta-learning:

- The learning agent continually adapts as it is evaluated on its predictions, rather than only adapting on $k$ labeled examples, as is common in few-shot learning.
- At train time, data is *unsegmented*, i.e. *not* grouped by the latent task $\mathcal{T}$.
- Similarly, at test time, the task changes with time, so the agent must infer which past data are drawn from the current task when making predictions.

Thus, the setting we consider here can be considered a generalization of the standard meta-learning setting, relaxing the requirement of task segmentation at train and test time. Both our problem setting and an illustration of the MOCA algorithm are presented in Fig. 1.

## 3 Preliminaries

**Meta-Learning**. The core idea of meta-learning is to directly optimize the few-shot learning performance of a machine learning model over a *distribution* of learning tasks, such that this learning performance generalizes to other tasks from this distribution.

A meta-learning method consists of two phases: meta-training and online adaptation. Let $\boldsymbol{\theta}$ be the parameters of this model learned in meta-training. During online adaptation, the model uses context data $\mathcal{D}_t = (\boldsymbol{x}_{1:t}, \boldsymbol{y}_{1:t})$ from within one task to compute statistics $\boldsymbol{\eta}_t = f_{\boldsymbol{\theta}}(\mathcal{D}_t)$, where $f$ is a function parameterized by $\boldsymbol{\theta}$. For example, in MAML [10], the statistics are the neural network weights after gradient updates computed using $\mathcal{D}_t$. For recurrent network-based meta-learning algorithms, these statistics correspond to the hidden state of the network. For a simple nearest-neighbors model, $\boldsymbol{\eta}$ may simply be the context data. The model then performs predictions by using these statistics to define a conditional distribution on $\boldsymbol{y}$ given new inputs $\boldsymbol{x}$, which we write $\boldsymbol{y} \mid \boldsymbol{x}, \mathcal{D}_t \sim p_{\boldsymbol{\theta}}(\boldsymbol{y} \mid \boldsymbol{x}, \boldsymbol{\eta}_t)$. Adopting a Bayesian perspective, we refer to $p_{\boldsymbol{\theta}}(\boldsymbol{y} \mid \boldsymbol{x}, \boldsymbol{\eta}_t)$ as the posterior predictive distribution. The performance of this model on this task can be evaluated through the log-likelihood of task data under this posterior predictive distribution $\mathcal{L}(\mathcal{D}_t, \boldsymbol{\theta}) = \mathbb{E}_{\boldsymbol{x}, \boldsymbol{y} \sim p(\cdot, \cdot | \mathcal{T}_i)}[- \log p_{\boldsymbol{\theta}}(\boldsymbol{y} \mid \boldsymbol{x}, f_{\boldsymbol{\theta}}(\mathcal{D}_t))]$.

Meta-learning algorithms, broadly, aim to optimize the parameters $\boldsymbol{\theta}$ such that the model performs well across a distribution of tasks, $\min_{\boldsymbol{\theta}} \ \mathbb{E}_{\mathcal{T}_i \sim p(\mathcal{T})}[\mathbb{E}_{\mathcal{D}_t \sim \mathcal{T}_i}[\mathcal{L}(\mathcal{D}_t, \boldsymbol{\theta})]]$. Across most meta-learning algorithms, both the update rule $f_{\boldsymbol{\theta}}(\cdot)$ and the prediction function are chosen to be differentiable operations, such that the parameters can be optimized via stochastic gradient descent. Given a dataset

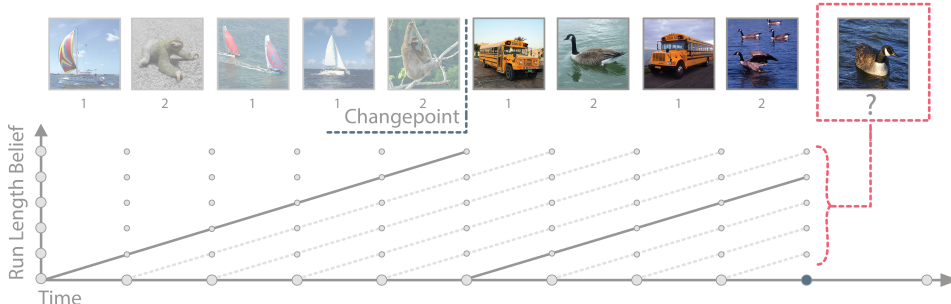

Figure 1: An illustration of a simplified version of our problem setting and of the MOCA algorithm. An agent sequentially observes an input $x$ (e.g, an image), makes a probabilistic prediction, and receives the true label $y$ (here, class 1 or 2). An unobserved change in the task (a "changepoint") results in a change in the generative model of $x$ and/or $y$. In the above image, the images corresponding to label 1 switch from sailboats to school buses, while the images corresponding to label 2 switch from sloths to geese. MOCA recursively estimates the time since the last changepoint, and conditions an underlying meta-learning model only on data that is relevant to the current task to optimize its predictions.

pre-segmented into groups of data from individual tasks, standard meta-learning algorithms can estimate this expectation by first sampling a group for which $\mathcal{T}$ is fixed, then treating one part as context data $\mathcal{D}_t$, and sampling from the remainder to obtain test points from the same task. While this strategy is effective for few-shot learning, it fails for settings like sequential prediction, where the latent task may change over time and segmenting data by task is difficult. Our goal is to bring meta-learning tools to such settings.

**Bayesian Online Changepoint Detection**. To enable meta-learning without task segmentation, we build upon Bayesian online changepoint detection [1], an approach for detecting discrete changes in a data stream (i.e. task switches), originally presented in an unconditional density estimation context.

BOCPD operates by maintaining a belief distribution over *run lengths*, i.e. how many past data points were generated under the current task. A run length $r_t = 0$ implies that the task has switched at time $t$, and so the current datapoint $y_t$ was drawn from a new task $\mathcal{T}' \sim p(\mathcal{T})$. We denote this belief distribution at time $t$ as $b_t(r_t) = p(r_t \mid y_{1:t-1})$. We can reason about the overall posterior predictive by marginalizing over the run length $r_t$ according to $b_t(r_t)$, $p(y_t \mid y_{1:t-1}) = \sum_{\tau=0}^{t-1} p(y_t \mid y_{1:t-1}, r_t = \tau) b_t(\tau)$, Given $r_t = \tau$, we know the past $\tau$ data points all correspond to the current task, so $p(y_t \mid y_{1:t-1}, r_t = \tau)$ can be computed as the posterior predictive of an underlying predictive model (UPM), conditioning on the past $\tau$ data points.

BOCPD recursively computes posterior predictive densities using this UPM for each value of $r_t \in \{0, \dots, t-1\}$, and then evaluates new datapoints $y_{t+1}$ under these posterior predictive densities to update the belief distribution $b(r_t)$. In this work, we extend these techniques to conditional density estimation, deriving update rules which use meta-learning models as the UPM.

## 4 Meta-Learning via Online Changepoint Analysis

We now present MOCA[1], which enables meta-learning in settings without task segmentation, both at train and test time. In the following subsections, we first extend BOCPD to derive a recursive Bayesian filtering algorithm for run length, leveraging a base meta-learning algorithm as the underlying predictive model (UPM). We then outline how the full framework allows both training and evaluating meta-learning models on time series without task segmentation.

### 4.1 Bayesian Task Duration Estimation

As in BOCPD, MOCA maintains a belief over possible run lengths $r_t$. Throughout this paper, we use $b_t$ to refer to the belief *before* observing data at that timestep, $(x_t, y_t)$. Note that $b_t$ is a discrete distribution with support over $r_t \in \{0, ..., t-1\}$. MOCA also maintains a version of the base meta-learning algorithm's posterior parameters $\eta$ for every possible run length. We write $\eta_t[r]$ to refer to the posterior parameters produced by the meta-learning algorithm after adapting to the past $r$

**Algorithm 1** Meta-Learning via Online Changepoint Analysis

---

**Require:** Training data $\boldsymbol{x}_{1:n}, \boldsymbol{y}_{1:n}$, number of training iterations $N$, initial model parameters $\boldsymbol{\theta}$
 1: **for** $i = 1$ to $N$ **do**
 2:     Sample training batch $\boldsymbol{x}_{1:T}, \boldsymbol{y}_{1:T}$ from the full timeseries.
 3:     Initialize run length belief $b_1(r_1 = 0) = 1$, posterior statistics $\boldsymbol{\eta}_0[r = 0]$ according to $\boldsymbol{\theta}$
 4:     **for** $t = 1$ to $T$ **do**
 5:        Observe $\boldsymbol{x}_t$, compute $b_t(r_t \mid \boldsymbol{x}_t)$ via (2)
 6:        Predict $p_{\boldsymbol{\theta}}(\hat{\boldsymbol{y}}_t \mid \boldsymbol{x}_{1:t}, \boldsymbol{y}_{1:t-1})$ via (5)
 7:        Observe $\boldsymbol{y}_t$ and incur NLL loss $\ell_t = -\log p_{\boldsymbol{\theta}}(\boldsymbol{y}_t \mid \boldsymbol{x}_{1:t}, \boldsymbol{y}_{1:t-1})$
 8:        Compute updated posteriors $\boldsymbol{\eta}_t[r_t]$ for all $r_t$ via (6)
 9:        Compute $b_t(r_t \mid \boldsymbol{x}_t, \boldsymbol{y}_t)$ via (3)
10:       Compute updated belief over run length $b_{t+1}$ via (4)
11:     **end for**
12:     Compute $\nabla_{\boldsymbol{\theta}} \sum_{t=k}^{k+T} \ell_t$ and take gradient descent step to update $\boldsymbol{\theta}$
13: **end for**

---

datapoints, $(\boldsymbol{x}_{t-r+1:t}, \boldsymbol{y}_{t-r+1:t})$. Given this collection of posteriors, we can compute the likelihood of observing data given the run length $r$. This allows us to apply rules from Bayesian filtering to update the run length belief in closed form. These updates involve three steps:

If the base meta-learning algorithm maintains a posterior distribution of inputs $p_{\boldsymbol{\theta}}(\boldsymbol{x}_t \mid \boldsymbol{\eta}_{t-1})$, then MOCA can update the belief $b_t$ directly after observing $\boldsymbol{x}_t$, as follows

$$b_t(r_t \mid \boldsymbol{x}_t) := p(r_t \mid \boldsymbol{x}_{1:t}, \boldsymbol{y}_{1:t-1}) \propto p_{\boldsymbol{\theta}}(\boldsymbol{x}_t \mid \boldsymbol{\eta}_{t-1}[r_t]) b_t(r_t) \qquad (2)$$

which can be normalized by summing over the finite support of $b_t$. This step relies on maintaining a generative model of the input variable, which is atypical for most regression models and is not done for discriminative classification models. While this filtering step is optional, it allows MOCA to detect task switches based on a changes in the input distribution when possible.

Next, upon observing the label $\boldsymbol{y}_t$, we can use the base meta-learning algorithm's conditional posterior predictive $p_{\boldsymbol{\theta}}(\boldsymbol{y}_t \mid \boldsymbol{x}_t, \boldsymbol{\eta}_{t-1})$ to again update the belief over run length:

$$b_t(r_t \mid \boldsymbol{x}_t, \boldsymbol{y}_t) := p(r_t \mid \boldsymbol{x}_{1:t}, \boldsymbol{y}_{1:t}) \propto p_{\boldsymbol{\theta}}(\boldsymbol{y}_t \mid \boldsymbol{x}_t, \boldsymbol{\eta}_{t-1}[r_t]) b_t(r_t \mid \boldsymbol{x}_t), \qquad (3)$$

which can similarly be normalized.

Finally, to push the run length belief forward in time, we note that we assume that the task switches with probability $\lambda$ at every timestep, and so the task remains fixed with probability $1 - \lambda$. This yields the update

$$b_{t+1}(r_{t+1} = k) = \begin{cases} \lambda & \text{if } k = 0 \\ (1 - \lambda) b_t(r_t = k - 1 \mid \boldsymbol{x}_t, \boldsymbol{y}_t) & \text{if } k > 0 \end{cases}. \qquad (4)$$

For more details on the derivation of these updates, we refer the reader to Appendix A.

## 4.2 Meta Learning without Task Segmentation

By taking a Bayesian filtering approach to changepoint detection, we avoid hard assignments of changepoints and instead perform a soft selection over run lengths. In this way, MOCA is able to backpropagate through the changepoint detection and directly optimize the underlying predictive model, which may be any meta-learning model that admits a probabilistic interpretation.

MOCA processes a time series sequentially. We initialize $b_1(r_1 = 0) = 1$, and initialize the posterior statistics for $\boldsymbol{\eta}_0[r_1 = 0]$ as specified by the parameters $\boldsymbol{\theta}$ of the meta learning algorithm. Then, at timestep $t$, we first observe inputs $\boldsymbol{x}_t$ and compute $b_t(r_t \mid \boldsymbol{x}_t)$ according to (2). Next, we marginalize to make a probabilistic prediction for the label, $p_{\boldsymbol{\theta}}(\hat{\boldsymbol{y}}_t \mid \boldsymbol{x}_{1:t}, \boldsymbol{y}_{1:t-1})$ equal to

$$\sum_{r_t=0}^{t-1} b_t(r_t \mid \boldsymbol{x}_t) p_{\boldsymbol{\theta}}(\hat{\boldsymbol{y}}_t \mid \boldsymbol{x}_t, \boldsymbol{\eta}_{t-1}[r_t]) \qquad (5)$$

We then observe the label $\boldsymbol{y}_t$ and incur the corresponding loss. We can also use the label both to compute $b_t(r_t \mid \boldsymbol{x}_t, \boldsymbol{y}_t)$ according to (3), as well as to update the posterior statistics for all the run

lengths using the labeled example. Many meta-learning algorithms admit a recursive update rule which allows these parameters to be computed efficiently using the past values of $\boldsymbol{\eta}$,

$$\boldsymbol{\eta}_t[r] = h(\boldsymbol{x}_t, \boldsymbol{y}_t, \boldsymbol{\eta}_{t-1}[r-1]) \quad \forall\ r = 1, \dots, t. \tag{6}$$

While MOCA could work without such a recursive update rule, this would require storing data online and running the non-recursive posterior computation $\boldsymbol{\eta}_t = f_{\boldsymbol{\theta}}((\boldsymbol{x}_{t-r_t+1:t}, \boldsymbol{y}_{t-r_t+1:t}))$ for every $r_t$, which involves $t$ operations using datasets of sizes from 0 to $t$, and thus can be an $O(t^2)$ operation. In contrast, the recursive updates involve $t$ operations involving just the latest datapoint, yielding $O(t)$ complexity. Finally, we propagate the belief over run length forward in time to obtain $b_t(r_{t+1})$ to be ready to process the next data point in the timeseries.

Since all these operations are differentiable, given a training time series in which there are task switches $\mathcal{D}_{\mathrm{train}}$, we can run this procedure, sum the negative log likelihood (NLL) losses incurred at each step, and use backpropagation within a standard automatic differentiation framework to optimize the parameters of the base learning algorithm, $\boldsymbol{\theta}$. Algorithm 1 outlines this training procedure. In practice, we sample shorter time series of length $T$ from the training data to ease computational requirements during training; we discuss implications of this in Appendix D. If available, a user can input various levels of knowledge on task segmentation by manually updating $b(r_t)$ at any time; further details and empirical validation of this task semi-segmented use case are also provided in Appendix D.

## 4.3 Making your MOCA: Model Instantiations

Thus far, we have presented MOCA at an abstract level, highlighting the fact that it can be used with any meta-learning model that admits the probabilistic interpretation as the UPM. Practically, as MOCA maintains several copies of the posterior statistics $\boldsymbol{\eta}$, meta-learning algorithms with lower-dimensional posterior statistics which admit recursive updates yield better computational efficiency. With this in mind, for our experiments we implemented MOCA using a variety of base meta-learners: an LSTM-based meta-learning approach [21], as well as meta-learning algorithms based on Bayesian modeling which exploit conjugate prior/likelihood models allowing for closed-form recursive posterior updates, specifically ALPaCA [16] for regression and a novel algorithm in a similar vein which we call PCOC, for probabilistic clustering for online classification, for classification. Further details on all methods are provided in Appendix B.

**LSTM Meta-learner.** The LSTM meta-learning approach encodes the information in the observed samples using hidden state $\boldsymbol{h}_t$ of an LSTM [20], and subsequently uses this hidden state to make predictions. Specifically, we follow the architecture proposed in [21], wherein an encoding of the current input $\boldsymbol{z}_t = \boldsymbol{\phi}(\boldsymbol{x}_t, \boldsymbol{w})$ as well as the previous label $\boldsymbol{y}_{t-1}$ are fed as input to the LSTM cell to update the hidden state $\boldsymbol{h}_t$ and cell state $\boldsymbol{c}_t$. For regression, the mean and variance of a Gaussian posterior predictive distribution are output as a function of the hidden state and encoded input $[\mu, \Sigma] = f(\boldsymbol{h}_t, \boldsymbol{z}_t; \boldsymbol{w}_f)$. The function $f$ is a feedforward network in both cases, with weights $\boldsymbol{w}_f$. Within the MOCA framework, the posterior statistics for this model are $\boldsymbol{\eta}_t = \{\boldsymbol{h}_t, \boldsymbol{c}_t, \boldsymbol{y}_t\}$.

**ALPaCA: Bayesian Meta-Learning for Regression.** ALPaCA is a meta-learning approach which performs Bayesian linear regression in a learned feature space, such that $\boldsymbol{y} \mid \boldsymbol{x} \sim \mathcal{N}(K^T \boldsymbol{\phi}(\boldsymbol{x}, \boldsymbol{w}), \Sigma_\epsilon)$ where $\boldsymbol{\phi}(\boldsymbol{x}, \boldsymbol{w})$ is a feed-forward neural network with weights $\boldsymbol{w}$ mapping inputs $\boldsymbol{x}$ to a $n_\phi$-dimensional feature space. ALPaCA maintains a matrix-normal distribution over $K$, and thus results in a matrix-normal posterior distribution over $K$. This posterior inference may be performed exactly, and computed recursively. The matrix-normal distribution on the last layer results in a Gaussian posterior predictive density. Note that, as is typical in regression, ALPaCA only models the conditional density $p(\boldsymbol{y} \mid \boldsymbol{x})$, and assumes that $p(\boldsymbol{x})$ is independent of the underlying task. The algorithm parameters $\boldsymbol{\theta}$ are the prior on the last layer, as well as the weights $\boldsymbol{w}$ of the neural network feature network $\boldsymbol{\phi}$. The posterior statistics $\boldsymbol{\eta}$ encode the mean and variance of the Gaussian posterior distribution on the last layer weights.

**PCOC: Bayesian Meta-Learning for Classification.** In the classification setting, one can obtain a similar Bayesian meta-learning algorithm by performing Gaussian discriminant analysis in a learned feature space. We refer to this novel approach to meta-learning for classification as probabilistic clustering for online classification (PCOC). Labeled input/class pairs $(\boldsymbol{x}_t, y_t)$ are processed by encoding the input through an embedding network $\boldsymbol{z}_t = \boldsymbol{\phi}(\boldsymbol{x}_t; \boldsymbol{w})$, and performing Bayesian density estimation in this feature space for every class. Specifically, we assume a Categorical-Gaussian generative model in this embedding space, and impose the conjugate Dirichlet prior over the class

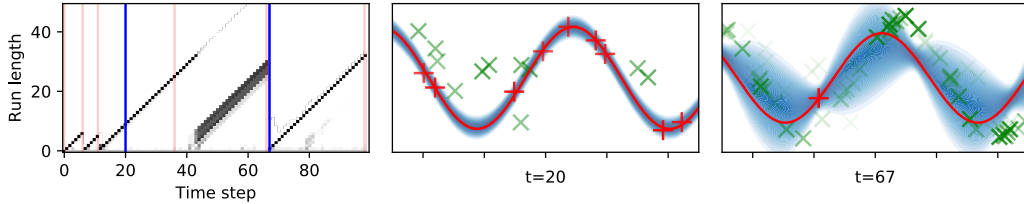

Figure 2: MOCA with ALPaCA on the sinusoid regression problem. **Left:** The belief over run length versus time. The intensity of each point in the plot corresponds to the belief in run length at the associated time. The red lines show the true changepoints. **Middle, Right:** Visualizations of the posterior predictive density at the times marked by blue lines in the left figure. The red line denotes the current function (task), and red points denote data from the current task. Green points denote data from previous tasks, where more faint points are older. By reasoning about task run-length, MOCA fits the current sinusoid while avoiding negative transfer from past data, and resets to prior predictions when tasks switch.

probabilities and a Gaussian prior over the mean for each class. This ensures the posterior remains Dirichlet-Gaussian, whose parameters can be updated recursively. The posterior parameters $\eta$ for this algorithm are the mean and covariance of the posterior distribution on each class mean, as well as the counts of observations per class. The learner parameters $\theta$ are the weights of the encoding network $w$, the prior parameters, and the covariance assumed for the observation noise. PCOC can be thought of a Bayesian analogue of prototypical networks [40].

# 5 Related Work

**Online Learning, Continuous Learning, and Concept Drift Adaptation.** A substantial literature exists on online, continual and lifelong learning [18, 6]. These fields all consider learning within a streaming series of tasks, wherein it is desirable to re-use information from previous tasks while avoiding negative transfer [12, 42]. Typically, continual learning assumes access to task segmentation information, whereas online learning does not [3]. Regularization approaches [26, 18, 28] have been shown to be an effective method for avoiding forgetting in continual learning. By augmenting the loss function for a new task with a penalty for deviation from the parameters learned for previous tasks, the regularizing effects of a prior are mimicked; in contrast we explicitly learn a prior over task weights that is meta-trained to be rapidly adaptive. Thus, MOCA is capable of avoiding substantial negative transfer by detecting task change, and rapidly adapting to new tasks. [3] loosen the assumption of task segmentation in continual learning and operate in a similar setting to that addressed herein, but they aim to optimize one model for all tasks simultaneously; in contrast, our work takes a meta-learning approach and aims to optimize a learning algorithm to quickly adapt to changing tasks.

**Meta-Learning for Continuous and Online Learning.** In response to the slow adaption of continual learning algorithms, there has been substantial interest in applying ideas from meta-learning to continual learning to enable rapid adaptation to new tasks. To handle streaming data, several works [31, 19] use a sliding window approach, wherein a fixed amount of past data is used to condition the meta-learned model. As this window length is not reactive to task change, these models risk suffering from negative transfer. Indeed, MOCA may be interpreted as sliding window model, that actively infers the optimal window length. [32] and [24] aim to detect task changes online by combining mean estimation of the labels with MAML. However, these models are less expressive than MOCA (which maintains a full Bayesian posterior), and require task segmentation as test time. [36] employ gradient-based meta-learning to improve transfer between tasks in continual learning; in contrast MOCA works with any meta-learning algorithm.

**Empirical Bayes for Changepoint Models.** Follow-on work to BOCPD [1] and the similar simultaneous work of [9] has considered applying empirical Bayes to optimize the underlying predictive model, a similar problem to that addressed herein. In particular, [33] develop a forward-backward algorithm that allows closed-form max likelihood estimation of the prior for simple distributions via EM. [43] derive general-purpose gradients for hyperparameter optimization within the BOCPD model. MOCA pairs these ideas with neural network meta-learning models, and thus can leverage recent advances in automatic differentiation for gradient computation.

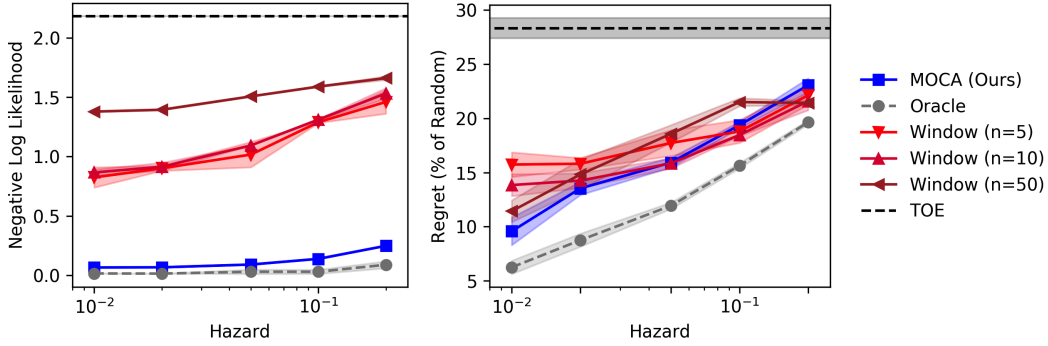

Figure 3: Performance of MOCA with ALPaCA versus baselines in sinusoid regression (**left**) and the switching wheel contextual bandit problem (**right**). In the bandit problem, we evaluate performance as the regret of the model (compared to an optimal decision maker with perfect knowledge of switch times) as a percentage of the regret of the random agent, following previous work [37]. In both problems, lower is better. Confidence intervals in this figure and throughout are 95%.

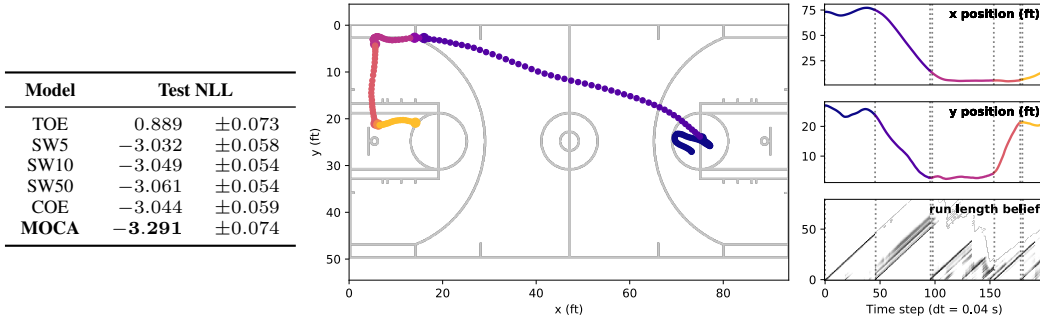

| Model | Test NLL | |
|-------|---------|--------|
| TOE | 0.889 | ±0.073 |
| SW5 | −3.032 | ±0.058 |
| SW10 | −3.049 | ±0.054 |
| SW50 | −3.061 | ±0.054 |
| COE | −3.044 | ±0.059 |
| **MOCA** | **−3.291** | ±0.074 |

Figure 4: **Left**: Test NLL of MOCA + LSTM against baselines. **Middle**: Visualization of sample trajectory, segmented by color according to predicted task changes. We see that task changes visually correspond to different plays. **Right**: Trajectories plotted against time, together with MOCA's belief over run length. Task switches (dashed gray) were placed where the MAP run length drops to a value less than 5.

## 6 Experimental Results

We investigate the performance of MOCA in five problem settings: three in regression and two in classification. Our primary goal is to characterize how effectively MOCA can enable meta-learning algorithms to perform without access to task segmentation. We compare against baseline sliding window models, which again use the same meta-learning algorithm, but always condition on the last $n$ data points, for $n \in \{5, 10, 50\}$. These baselines are a competitive approach to learning in time-varying data streams [13] and have been applied meta-learning in time-varying settings [31]. We also compare to a "train on everything" model, which only learns a prior and does not adapt online, corresponding to a standard supervised learning approach. Finally, where possible, we compare MOCA against an "oracle" model that uses the same base meta-learning algorithm, but has access to exact task segmentation at train and test time, to explicitly characterize the utility of task segmentation. Due to space constraints, this section contains only core numerical results for each problem setting; further experiments and ablations are presented in the appendix. We find by explicitly reasoning about task run-length, MOCA is able to outperform baselines across all the domains with a variety of base meta-learning algorithms and provide interpretable estimates of task-switches at test time.

**Sinusoid Regression**. To characterize MOCA in the regression setting, we investigate the performance on a switching sinusoid problem adapted from [10], in which a task change corresponds to a re-sampled sinusoid phase and amplitude. Qualitative results are visualized for the sinusoid in Fig. 2. In this problem we pair MOCA with ALPaCA as it outperforms LSTM-based meta-learners. MOCA is capable of accurate and calibrated posterior inference with only a handful of data points, and is capable of rapidly identifying task change. Typically, it identifies task change in one timestep, unless the datapoint happens to have high likelihood under the previous task as in Fig. 2d. Performance of MOCA against baselines is presented in Fig. 3 for all problem domains. For sinusoid (left), MOCA

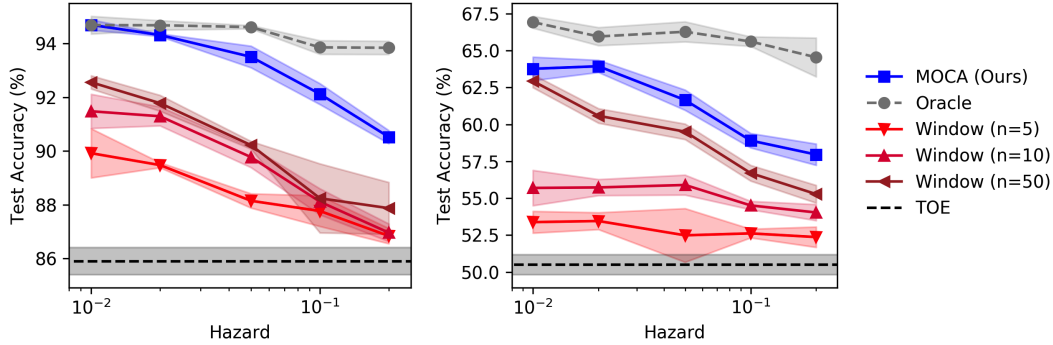

Figure 5: Performance of MOCA with PCOC on rainbow MNIST (**left**) and miniImageNet (**right**). In both problems, higher is better.

achieves performance close to the oracle model and substantially outperforms the sliding window approaches for all hazard rates.

**Wheel Bandit**. Bandit problems have seen recent highly fruitful application of meta-learning algorithms [4, 45, 15]. We investigate the performance of MOCA (paired with ALPaCA) in the *switching* bandit problem, in which the reward function of the bandit undergoes discrete changes [14, 17, 30]. We extend the wheel bandit problem [37], a common benchmark for meta-learning algorithms [15, 34]. Details of the full bandit problem are provided in the appendix. In this problem, changepoint identification is difficult, as only a small subset of states contains information about whether the reward function has changes.

Following [30], we use Thompson sampling for action selection. We use the notion of regret defined in [14], in which the chosen action is compared to the action with the best mean reward at each time, with perfect knowledge of switches. As shown in [14], the sliding window baselines have strong theoretical guarantees on regret, as well as good empirical performance. Performance is plotted in Fig. 3. MOCA outperforms baselines for lower hazard rates. Detecting task switches requires observing a state close to the (changing) high-reward boundary, and at high hazard rates, the rapid task changes make identification of changepoints difficult, and we see that MOCA performance matches all the sliding windows in this regime.

**NBA Player Movement**. To test MOCA on a real-world data with an unobserved switching latent task, we test it on predicting the movement of NBA players, whose intent may switch over time, from, e.g., running towards a position on the three-point line, to moving inside the key to recover a rebound. This changing latent state has made it a common benchmark for recurrent predictive models [22, 29]. In our experiments, the input $x$ is an individual player's current position on the court $(x_t, y_t)$, and the label $y_t = x_{t+1} - x_t$ is the step the player takes at that time. For this problem, we pair MOCA with the LSTM meta-learner, since recurrent models are well suited to this task and we saw better performance relative to ALPaCA. We add a "condition on everything" (COE) baseline which updates a single set of posterior statistics $\eta$ using all available data, as the LSTM can theoretically learn to only consider relevant data. Nevertheless, we find that that MOCA's explicit reasoning over task length yields better performance over COE and other baselines, as shown in Fig. 4. While true task segmentation is unavailable for this data, we see in the figure that MOCA's predictions of task changes correspond intuitively to changes in the player's intent.

**Rainbow MNIST**. In the classification setting, we apply MOCA with PCOC to the Rainbow MNIST dataset of [11]. In this dataset, MNIST digits have been perturbed via a color change, rotation, and scaling; each task corresponds to a unique combination of these transformations. Relative to baselines, MOCA approaches oracle performance for low hazard rates, due in part to the fact that task change can usually be detected prior to prediction via a change in digit color. Seven colors were used, so with probability $6/7$, MOCA has a strong indicator of task change before observing the image class.

**miniImageNet**. Finally, we investigate the performance of MOCA with PCOC on the miniImageNet benchmark task [44]. This dataset consists of 100 ImageNet categories [7], each with 600 RGB images of resolution $84 \times 84$. In our continual learning setting, we associate each class with a semantic label that is consistent between tasks. As five-way classification is standard for miniImageNet [44, 40], we split the miniImageNet dataset in to five approximately balanced "super-classes." For example,

one super-class is dog breeds, while another is food, kitchen and clothing items; details are provided in the appendix. Each new task corresponds to resampling a particular class from each super-class from which to draw inputs $x$; the labels $y$ remain the five super-classes, enabling knowledge re-use between classes. This corresponds to a continual learning scenario in which each super-class experiences distributional shift over time. Fig. 5 shows that MOCA outperforms baselines for all hazard rates.

## 7 Discussion and Conclusions

**Future Work.** In this work, we address the case in which tasks are sampled i.i.d. from a (typically continuous) distribution, and thus knowledge re-use adds marginal value. However, many domains may have tasks that can reoccur, or temporal dynamics to task evolution and thus data efficiency may be improved by re-using information for previous tasks. Previous work [32, 24, 27] has addressed the case in which tasks reoccur in both meta-learning and the BOCPD framework, and thus knowledge (in the form of a posterior estimate) may be re-used. Broadly, moving beyond the assumption of i.i.d. tasks to tasks having associated dynamics [2] represents a promising future direction.

**Conclusions.** MOCA enables the application of existing meta-learning algorithms to problems without task segmentation, such as the problem setting of continual learning. We find that by leveraging a Bayesian perspective on meta-learning algorithms and augmenting these algorithms with a Bayesian changepoint detection scheme to automatically detect task switches within time series, we can achieve similar predictive performance when compared to the standard task-segmented meta-learning setting, without the often prohibitive requirement of supervised task segmentation.

## Funding Disclosure and Acknowledgments

James Harrison was supported in part by the Stanford Graduate Fellowship and the National Sciences and Engineering Research Council of Canada (NSERC). The authors were partially supported by an Early Stage Innovations grant from NASA's Space Technology Research Grants Program, and by DARPA, Assured Autonomy program. The authors wish to thank Matteo Zallio for help in the design of figures.

## Broader Impact

Our work provides a method to extend meta-learning algorithms beyond the task-segmented case, to the time series series domain. Equivalently, our work extends core methods in changepoint detection, enabling the use of highly expressive predictive models via empirical Bayes. This work has the potential to extend the domain of applicability of both of these methods. Standard meta-learning relies on a collection of datasets, each corresponding to discrete tasks. A natural question is how such datasets are constructed; in many cases, these datasets rely on segmentation of time series data by experts. Thus, our work has the potential to make meta-learning algorithms applicable to problems that, previously, would have been too expensive or impossible to segment. Moreover, our work has the potential to improve the applicability of changepoint detection methods to difficult time series forecasting problems.

While MOCA has the potential to expand the domain of problems addressable via meta-learning, this has the effect of amplifying the risks associated with these methods. Meta-learning enables efficient learning for individual members of a population via leveraging empirical priors. There are clear risks in few-shot learning generally: for example, efficient facial recognition from a handful of images has clear negative implications for privacy. Moreover, while there is promising initial work on fairness for meta-learning [39], we believe considerable future research is required to understand the degree to which meta-learning algorithms increase undesirable bias or decrease fairness. While it is plausible that fine-tuning to the individual results in reduced bias, there are potential unforeseen risks associated with the adaptation process, and future research should address how bias is potentially introduced in this process. Relative to decision making rules that are fixed across a population, algorithms which fine-tune decision making to the individual present unique challenges in analyzing fairness. Further research is required to ensure that the adaptive learning enabled by algorithms such as MOCA do not lead to unfair outcomes.

## Footnotes

[1]Code is available at https://github.com/StanfordASL/moca

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
