[Supplementary Material]

# A  MOCA Algorithmic Details

In this section, we derive the Bayesian belief updates used in MOCA. As in the paper, we will use $b_t$ to refer to the belief *before* observing data at that timestep, $(\boldsymbol{x}_t, \boldsymbol{y}_t)$. Note that $b_t$ is a discrete distribution with support over $r_t \in \{0, ..., t-1\}$. We write $\boldsymbol{\eta}_t[r]$ to refer to the posterior parameters of the meta-learning algorithm conditioned on the past $r$ data points, $(\boldsymbol{x}_{t-r+1:t}, \boldsymbol{y}_{t-r+1:t})$.

At time $t$, the agent first observes the input $\boldsymbol{x}_t$, then makes a prediction $p(\boldsymbol{y}_t \mid \boldsymbol{x}_{1:t}, \boldsymbol{y}_{1:t-1})$, and subsequently observes $\boldsymbol{y}_t$. Generally, the latent task can influence both the marginal distribution of the input, $p(\boldsymbol{x}_t \mid \boldsymbol{x}_{1:t-1}, \boldsymbol{y}_{1:t-1})$ as well as the conditional distribution $p(\boldsymbol{y}_t \mid \boldsymbol{x}_{1:t}, \boldsymbol{y}_{1:t-1})$. Thus, the agent can update its belief over run lengths once after observing the input $\boldsymbol{x}_t$, and again after observing the label $\boldsymbol{y}_t$. We will use $b_t(r_t \mid \boldsymbol{x}_t) = p(r_t \mid \boldsymbol{x}_{1:t}, \boldsymbol{y}_{1:t-1})$ to represent the updated belief over run length after observing only $\boldsymbol{x}_t$, and $b_t(r_t \mid \boldsymbol{x}_t, \boldsymbol{y}_t) = p(r_t \mid \boldsymbol{x}_{1:t}, \boldsymbol{y}_{1:t})$ to represent the fully updated belief over $r_t$ after observing $\boldsymbol{y}_t$. Finally, we will propagate this forward in time according to our assumptions on task dynamics to compute $b_{t+1}(r_{t+1})$, which is used in the subsequent timestep.

To derive the Bayesian update rules, we start by noting that the updated posterior is proportional to the joint density,

$$b_t(r_t \mid \boldsymbol{x}_t) = p(r_t \mid \boldsymbol{x}_{1:t}, \boldsymbol{y}_{1:t-1}) \tag{7}$$
$$= Z^{-1} p(r_t, \boldsymbol{x}_t \mid \boldsymbol{x}_{1:t-1}, \boldsymbol{y}_{1:t-1})$$
$$= Z^{-1} p(\boldsymbol{x}_t \mid \boldsymbol{x}_{1:t-1}, \boldsymbol{y}_{1:t-1}, r_t) b_t(r_t) \tag{8}$$

where the normalization constant $Z$ can be computed by summing over the finite support of $b_{t-1}(r_t)$. Importantly, this update requires $p_{\boldsymbol{\theta}}(\boldsymbol{x}_t \mid \boldsymbol{\eta}_{t-1}[r_t])$, the base meta-learning algorithm's posterior predictive density over the inputs. Within classification, this density is available for generative models, and thus a generative approach is favorable to a discriminative approach within MOCA. In regression, it is uncommon to estimate the distribution of the independent variable. We take the same approach in this work and assume that $\boldsymbol{x}_t$ is independent of the task for regression problems, in which case $b_t(r_t \mid \boldsymbol{x}_t) = b_t(r_t)$.

Next, upon observing $\boldsymbol{y}_t$, we can similarly factor the belief over run lengths for the next timestep,

$$b_t(r_t \mid \boldsymbol{x}_t, \boldsymbol{y}_t) \propto p_{\boldsymbol{\theta}}(\boldsymbol{y}_t \mid \boldsymbol{x}_t, \boldsymbol{\eta}_{t-1}[r_t]) b_t(r_t \mid \boldsymbol{x}_t), \tag{9}$$

which can again easily be normalized.

Finally, we must propagate this belief forward in time:

$$b_{t+1}(r_{t+1}) = p(r_{t+1} \mid \boldsymbol{x}_{1:t}, \boldsymbol{y}_{1:t})$$
$$= \sum_{r_t} p(r_{t+1}, r_t \mid \boldsymbol{x}_{1:t}, \boldsymbol{y}_{1:t})$$
$$= \sum_{r_t} p(r_{t+1} \mid r_t) b_t(r_t \mid \boldsymbol{x}_t, \boldsymbol{y}_t).$$

where we have exploited the assumption that the changes in task, and hence the evolution of run length $r_t$, happen independently of the data generation process. The conditional run-length distribution $p(r_{t+1} \mid r_t)$ is defined by our model of task evolution.

Recall that we assume that the task switches with fixed probability $\lambda$, the hazard rate. Thus, $p(r_{t+1} = 0 \mid r_t) = \lambda$ for all $r_t$, implying $b_{t+1}(r_{t+1} = 0) = \lambda$. Conditioned on the task remaining the same, $r_{t+1} = k > 0$ and $r_t = k - 1$. Thus, $p(r_{t+1} = k \mid r_t) = (1 - \lambda)\mathbb{1}\{r_t = k - 1\}$ implying

$$b_{t+1}(r_{t+1} = k) = (1 - \lambda)b_t(r_t = k - 1 \mid \boldsymbol{x}_t, \boldsymbol{y}_t). \tag{10}$$

This gives the time-propagation update step, as in equation (4), used by MOCA.

# B  Base Meta-Learning Algorithm Details

In the following subsections, we describe how each of the base meta-learning algorithms we use for the experiments fit into the MOCA framework. Specifically, we highlight (1) which parameters $\boldsymbol{\theta}$ are optimized, (2) the statistics $\boldsymbol{\eta}$ for each algorithm, (3) how these statistics define a posterior predictive distribution $p_{\boldsymbol{\theta}}(\hat{\boldsymbol{y}}_{t+1} \mid \boldsymbol{x}_{1:t+1}, \boldsymbol{y}_{1:t})$, and finally (4) the recursive update rule $\boldsymbol{\eta}_t = h(\boldsymbol{\eta}_{t-1}, \boldsymbol{x}_t, \boldsymbol{y}_t)$ used to incorporate a new labeled example.

## B.1 LSTM Meta-Learner

For our LSTM meta-learner, we follow the architecture of [21]. The LSTM input is the concatenation of the current encoded input $z_t = \phi(x_t, w)$ and the label from the past timestep $y_{t-1}$. In this way, through the LSTM update process, the hidden state can process a sequence of input/label pairs and encode statistics of the posterior distribution in the hidden state. Thus, the necessary statistics to make predictions after observing $x_{1:t}$ and $y_{1:t}$ are $\eta_t = [h_t, c_t, y_t]$. Given a new example $x, y$, and the posterior at time $t$, the updated posterior can be computed recursively

$$h_{t+1}, c_{t+1} = \text{LSTM}([x, y_t], h_t, c_t) \tag{11}$$

$$y_{t+1} = y \tag{12}$$

where $\text{LSTM}([x, y_t], h, c)$ carries out the LSTM update rules for hidden and cell states given input $[x, y_t]$.

We depart from the architecture proposed in [21] and include both the hidden state $h_t$ and the current encoded input $z_t$ as input to the decoder $f$ which outputs the statistics of the posterior predictive distribution $\hat{y}_t \sim \mathcal{N}(\mu_t, \Sigma_t)$.

$$\mu_t, s_t = f(h_t, z_t, w_f) \tag{13}$$

$$\Sigma = \mathbf{diag}(\exp(s_t)) \tag{14}$$

where $f$ is a single hidden layer feed-forward network with weights $w_f$. This functional form ensures that the covariance matrix of the posterior predictive remains positive definite. By including $z_t$ as input to the decoder, we lessen the information that needs to be stored in the hidden state, as it no longer needs to also encode the posterior predictive density for $y \mid x_t$, just the posterior on the latent task. This was found to substantially improve performance and learning stability.

The parameters that are optimized during meta-training are the weights of the encoder and decoder $w, w_f$, as well as the parameters of the LSTM gates.

The LSTM meta-learner makes few assumptions on the structure of the probabilistic model of the unobserved task parameter. For example, it does not by design satisfy the exchangeability criterion ensuring that the order of the context data does not change the posterior. This makes it a flexible algorithm that, e.g., can handle unobserved latent states that have dynamics (both slow varying and switching behavior, in theory). However, empirically we find the lack of this structure can make these models harder to train. Indeed, the more structured algorithms introduced in the following sections outperformed the LSTM meta-learner on many of our experiments.

## B.2 ALPaCA

ALPaCA [16] is a meta-learning approach for which the base learning model is Bayesian linear regression in a learned feature space, such that $y \mid x \sim \mathcal{N}(K^T \phi(x, w), \Sigma_\epsilon)$.

We fix the prior $K \sim \mathcal{MN}(\bar{K}_0, \Sigma_\epsilon, \Lambda_0^{-1})$. In this matrix-normal prior, $\bar{K}_0 \in \mathbb{R}^{n_\phi \times n_y}$ is the prior mean and $\Lambda_0$ is a $n_\phi \times n_\phi$ precision matrix (inverse of the covariance). Given this prior and data model, the posterior may be recursively computed as follows. First, we define $Q_t = \Lambda_t^{-1} \bar{K}_t$. Then, the one step posterior update is

$$\Lambda_{t+1}^{-1} = \Lambda_t^{-1} - \frac{(\Lambda_t^{-1} \phi(x_{t+1}))(\Lambda_t^{-1} \phi(x_{t+1}))^T}{1 + \phi^T(x_{t+1})\Lambda_t^{-1}\phi(x_{t+1})}, \tag{15}$$

$$Q_{t+1} = y_{t+1}\phi^T(x_{t+1}) + Q_t \tag{16}$$

and the posterior predictive distribution is

$$p_\theta(\hat{y}_{t+1} \mid x_{1:t+1}, y_{1:t}) = \mathcal{N}(\mu(x_{t+1}), \Sigma(x_{t+1})), \tag{17}$$

where $\mu(x_{t+1}) = (\Lambda_t^{-1} Q_t)^T \phi(x_{t+1})$ and

$$\Sigma(x_{t+1}) = (1 + \phi^T(x_{t+1})\Lambda_t^{-1}\phi(x_{t+1}))\Sigma_\epsilon.$$

To summarize, ALPaCA is a meta learning model for which the posterior statistics are $\eta_t = \{Q_t, \Lambda_t^{-1}\}$, and the recursive update rule $h(x, y, \eta)$ is given by (16). The parameters that are meta-learned are the prior statistics, the feature network weights, and the noise covariance: $\theta = \{\bar{K}_0, \Lambda_0, w, \Sigma_\epsilon\}$. Note that, as is typical in regression, ALPaCA only models the conditional density $p(y \mid x)$, assuming that $p(x)$ is independent of the underlying task.

## B.3 PCOC

In PCOC we process labeled input/class pairs $(\boldsymbol{x}_t, y_t)$ by encoding the input through an embedding network $\boldsymbol{z}_t = \boldsymbol{\phi}(\boldsymbol{x}_t; \boldsymbol{w})$, and performing Bayesian density estimation for every class. Specifically, we assume a Categorical-Gaussian generative model in this embedding space, and impose the conjugate Dirichlet prior over the class probabilities and a Gaussian prior over the mean for each class,

$$y_t \sim \operatorname{Cat}(p_1, \ldots, p_{n_y}), \quad p_1, \ldots, p_{n_y} \sim \operatorname{Dir}(\boldsymbol{\alpha}_0),$$
$$\boldsymbol{z}_t \mid y_t \sim \mathcal{N}(\bar{z}_{y_t}, \Sigma_{\epsilon, y_t}), \quad \bar{z}_{y_t} \sim \mathcal{N}(\mu_{y_t, 0}, \Lambda_{y_t, 0}^{-1}).$$

Given labeled context data $(\boldsymbol{x}_t, y_t)$, the algorithm updates its belief over the Gaussian mean for the corresponding class, as well as its belief over the probability of each class. As with ALPaCA, these posterior computations can be performed through closed form recursive updates. Defining $\boldsymbol{q}_{i,t} = \Lambda_{i,t} \boldsymbol{\mu}_{i,t}$, we have

$$\boldsymbol{\alpha}_t = \boldsymbol{\alpha}_{t-1} + \mathbf{1}_{y_t}, \quad \boldsymbol{q}_{y_t,t} = \boldsymbol{q}_{y_t,t-1} + \Sigma_{\epsilon, y_t}^{-1} \boldsymbol{\phi}(\boldsymbol{x}_t), \quad \Lambda_{y_t,t} = \Lambda_{y_t,t-1} + \Sigma_{\epsilon, y_t}^{-1} \quad (18)$$

where $\mathbf{1}_i$ denotes a one-hot vector with a one at index $i$. Terms not related to class $y_t$ are left unchanged in this recursive update. Given this set of posterior parameters $\boldsymbol{\eta}_t = \{\boldsymbol{\alpha}_t, \boldsymbol{q}_{1:J,t}, \Lambda_{1:J,t}\}$, the posterior predictive density in the embedding space can be computed as

$$p(y \mid \boldsymbol{\eta}_t) = \alpha_{y,t} / (\sum_{i=1}^J \alpha_{i,t})$$
$$p(\boldsymbol{z}, y \mid \boldsymbol{\eta}_t) = p(y \mid \boldsymbol{\eta}_t) \mathcal{N}(\boldsymbol{z}; \Lambda_{y,t}^{-1} \boldsymbol{q}_{y,t}, \Lambda_{y,t}^{-1} + \Sigma_{\epsilon, y})$$

where $\mathcal{N}(\boldsymbol{z}; \mu, \Sigma)$ denotes the Gaussian pdf with mean $\mu$ and covariance $\Sigma$ evaluated at $\boldsymbol{z}$. Applying Bayes rule, the posterior predictive on $y_{t+1}$ given $\boldsymbol{z}_{t+1}$ is

$$p(\hat{y} \mid \boldsymbol{x}_{1:t+1}, \boldsymbol{y}_{1:t}) = \frac{p(\boldsymbol{z}_{t+1}, \hat{y} \mid \boldsymbol{\eta}_t)}{\sum_{y'} p(\boldsymbol{z}_{t+1}, y' \mid \boldsymbol{\eta}_t)}, \quad (19)$$

where $\boldsymbol{z}_{t+1} = \boldsymbol{\phi}(\boldsymbol{x}_{t+1})$. This generative modeling approach also allows computing $p(\boldsymbol{z}_{t+1} \mid \boldsymbol{\eta}_t)$ by simply marginalizing out $y$ from the joint density of $p(\boldsymbol{z}, y)$,

$$p(\boldsymbol{z}_{t+1} \mid \boldsymbol{\eta}_t) = \sum_{y=1}^J p(y) \mathcal{N}(\boldsymbol{z}_{t+1}; \mu_t, \Lambda_{y,t}^{-1} + \Sigma_{\epsilon, y})$$

As this only depends on the input $\boldsymbol{x}$, we can use this likelihood within MOCA to update the run length belief upon seeing $\boldsymbol{x}_t$ and before predicting $\hat{y}_t$.

In summary, PCOC leverages Bayesian Gaussian discriminant analysis, meta-learning the parameters $\boldsymbol{\theta} = \{\boldsymbol{\alpha}_0, \boldsymbol{q}_{1:J,0}, \Lambda_{1:J,0}, \boldsymbol{w}, \Sigma_{\epsilon, 1:J}\}$ for efficient few-shot online classification. In practice, we assume that all the covariances are diagonal to limit memory footprint of the posterior parameters.

**Discussion**. PCOC extends a line of work on meta-classification based on prototypical networks [40]. This framework maps the context data to an embedding space, after which it computes the centroid for each class. For a new data point, it models the probability of belonging to each class as the softmax of the distances between the embedded point and the class centroids, for some distance metric. For Euclidean distances (which the authors focus on), this corresponds to performing frequentist estimation of class means, under the assumption that the variance matrix for each class is the identity matrix[2]. Indeed, this corresponds to the cheapest-to-evaluate simplification of PCOC. [35] propose adding a class-dependent length scale (which is a scalar), which corresponds to meta-learning a frequentist estimate of the variance for each class. Moreover, it corresponds to assuming a variance that takes the form of a scaled identity matrix. Indeed, assuming diagonality of the covariance matrix results in substantial performance improvement as the matrix inverse may be performed element-wise. This reduces the numerical complexity of this operation in the (frequently high-dimensional) embedding space from cubic to linear. In our implementation of MOCA, we assume diagonal covariances throughout, resulting in comparable computational complexity to the different flavors of prototypical networks. If one were to use dense covariances, the computational performance decreases substantially (due to the necessity of expensive matrix inversions), especially in high dimensional embedding spaces.

In contrast to this previous work, PCOC has several desirable features. First, both [40] and [35] make the implicit assumption that the classes are balanced, whereas we perform online estimation of class probabilities via Dirichlet posterior inference. Beyond this, our approach is explicitly Bayesian, and we maintain priors over the parameters that we estimate online. This is critical for utilization in the MOCA framework. Existence of these priors allows "zero-shot" learning—it enables a model to classify incoming data to a certain class, even if no data belonging to that class has been observed within the current task. Finally, because the posteriors concentrate (the predictive variance decreases as more data is observed), we may better estimate when a change in the task has occurred. We also note that maximum likelihood estimation of Gaussian means is dominated by the James-Stein estimator [41], which shrinks the least squares estimator toward some prior. Moreover, the James-Stein estimator paired with empirical Bayesian estimation of the prior—which is the basis for Bayesian meta-learning approaches such as ALPaCA and PCOC—has been shown to be a very effective estimator in this problem setting [8].

## C   Experimental Details

### C.1   Problem Settings

**Sinusoid**. To test the performance of the MOCA framework combined with ALPaCA for the regression setting, we investigate a switching sinusoid regression problem. The standard sinusoid regression problem, in which randomly sampled phase and amplitude constitute a task, is a standard benchmark in meta-learning [10]. Moreover, a switching sinusoid problem is a popular benchmark in continuous learning [19, 23]. Each task consists of a randomly sampled phase in the range $[0, \pi]$ and amplitude in $[0.1, 5]$. This task was investigated for varying hazard rates. For the experiments in this paper, samples from the sinusoid had additive zero-mean Gaussian noise of variance 0.05.

**Wheel Bandit**. As a second, more practical regression example, we investigate a modified version of the wheel bandit presented in [37]. This bandit has been used to evaluate several Bayesian meta-learning algorithms [15, 34], due to the fact that the problem requires effective exploration (which itself relies on an accurate model of the posterior). We will outline the standard problem, and then discuss our modified version.

The wheel problem is a contextual bandit problem in which a state $\boldsymbol{x} = [x_1, x_2]^T$ is sampled uniformly from the unit ball. The unit ball is split into two regions according to a radius $\delta \in [0, 1]$, and into four quadrants (for details, see [37]). There are five actions, $a_0, \ldots, a_4$. The first, $a_0$ always results in reward $r_m$. The other four actions each have one associated quadrant. For state $\boldsymbol{x}$ in quadrant 1, with $\|\boldsymbol{x}\| > \delta$, $a_1$ returns $r_h$, and all other actions return reward $r_l$. Actions $a_2, a_3, a_4$ all return $r_l$. If $\|\boldsymbol{x}\| \leq \delta$, $a_1$ also returns $r_l$. In quadratic 2, $a_2$ returns $r_h$ for $\boldsymbol{x} > \delta$, and so on. Critically, $\mathbb{E}[r_l] < \mathbb{E}[r_m] < \mathbb{E}[r_h]$. In summary, $a_0$ always returns a medium reward, whereas actions $a_1, \ldots, a_4$ return high reward for the correct quadrant outside of the (unknown) radius $\delta$, and otherwise return low reward.

We make several modifications to the setting to be better suited to the switching bandit setting. The standard wheel bandit problem is focused on exploration over long horizons. In the standard problem, the radius of the wheel is fixed, and an algorithm must both learn the structure of the problem and infer the radius. In meta-learning-based investigations of the problem, a collection of wheel bandit problems with different radii are provided for training. Then, at test time, a new problem with a previously unseen radius is provided, an the decision-making agent must correctly infer the radius. In our switching setting, the radius of the wheel changes sharply, randomly in time. The radius was sampled $\delta \sim \mathcal{U}[0, 1]$ in previous work [15, 34]. In our setting, with probability $\lambda$ at each time step (the hazard), the radius is re-sampled from this uniform distribution. Thus, the agent must constantly be inferring the current radius. Note that in this problem, only a small subset of states allow for meaningful exploration. Indeed, if the problem switches from radius $\delta_1$ to $\delta_2$, only $\boldsymbol{x}$ such that $\|\boldsymbol{x}\| \in [\delta_1, \delta_2]$ provides information about the switch. Thus, this problem provides an interesting domain in which changepoint detection is difficult and necessarily temporally delayed.

In addition to changing the sampling of the radius, we also change the reward function. As in [37], the rewards are defined as $r_i \sim \mathcal{N}(\mu_i, \sigma^2)$ for $i = l, m, h$. In [37, 15, 34], $\mu_l = 1.0$, $\mu_m = 1.2$, and $\mu_3 = 50.0$; $\sigma = 0.01$. This reward design results in agents necessarily needing to accurately identifying the radius of the problem, as for states outside of this value they may take the high reward action, and otherwise the agent takes action $a_0$, resulting in reward of (approximately) 1.2. While this results in an interesting exploration versus exploitaion problems in the long horizon, the relatively

greedy strategy of always choosing the action corresponding the quadrant of the state (potentially yielding high reward) performs well over short horizons. Thus, we modified the reward structure to make the shorter horizon problem associated with the switching bandit more interesting. In particular, we set $\mu_l = 0.0$, $\mu_m = 1.0$, $\mu_3 = 2.0$ and $\sigma = 0.5$. Thus, while the long horizon exploration problem is less interesting, a greedy agent performs worse over the short horizon. Moreover, the substantially higher noise variance increases the difficulty of the radius inference problem as well as the changepoint inference problem.

**NBA Player Movement**. The behavior of basketball players is well described as a sequence of distinct plays ("tasks"), e.g. running across the court or driving in towards the basket. As such, predicting a player's movement requires To generate data, we extracted $8$ second trajectories of player movement sampled at 12.5 Hz from games from the 2015-2016 NBA season[3]. For the training data, we used trajectories from two games randomly sampled from the dataset: the November 11th, 2015 game between the Orlando Magic and the Chicago Bulls, and the December 12, 2015 game between the New Orleans Pelicans and the Chicago Bulls. The validation data was extracted from the November 7th, 2015 game between the New Orleans Pelicans and the Dallas Mavericks. The test set was trajectories from the November 6th game between the Milwaukee Bucks and the New York Knicks. The input $\boldsymbol{x}_t$ was the player's $(x, y)$ position at time $t$, scaled down by a factor of 50. The labels were the unscaled changes in position, $\boldsymbol{y}_t = 50(\boldsymbol{x}_{t+1} - \boldsymbol{x}_t)$. The scaling was performed to convert the inputs, with units of feet and taking on values ranging from 0-100, to values that are more amenable for training with standard network initialization.

**Rainbow MNIST**. The Rainbow MNIST dataset (introduced in [11]) contains 56 different color/scale/rotation transformations of the MNIST dataset, where one transformation constitutes a task. We split this dataset into a train set of 49 transformations and a test set of 7. For hyperparameter optimization, we split the train set into a training set of 42 transformations and a validation of 7. However, because the dataset represents a fairly small amount of tasks (relative to the sinusoid problem, which has infinite), after hyperparameters were set we trained on all 49 tasks. We found this notably improved performance. Note that the same approach was used in [40].

**miniImageNet**. We use the miniImageNet dataset of [44], a standard benchmark in few-shot learning. However, the standard few-shot learning problem does not require data points to be assigned to a certain class label. Instead, given context data, the goal is to associated the test data with the correct context data. We argue that this problem setting is implausible for the continual learning setting: while observing a data stream, you are also inferring the set of possible labels. Moreover, after a task change, there is no context data to associate a new point with. Therefore we instead assume a known set of classes. We group the 100 classes of miniImageNet in to five super-classes, and perform five-way classification given these. These super-classes vary in intra-class diversity of sub-classes: for example, one of the super-class is entirely composed of sub-classes that are breeds of dogs, while another corresponds to buildings, furniture, and household objects. Thus, the strength of the prior information for each super-class varies. Moreover, the intra-class similarities are quite weak, and thus generalization from the train set to the test set is difficult and few-shot learning is still necessary and beneficial. The super-classes are detailed in table 1.

The super-classes are roughly balanced in terms of number of classes contained. Each task correspond to sampling a class from within each super-class, which was fixed for the duration of that task. Each super-class was sampled with equal probability.

## C.2 Baselines

Four baselines were used, described below:

- **Train on Everything**: This baseline consists of ignoring task variation and treating the training timeseries as one dataset. Note that many datasets contain latent temporal information that is ignored, and so this approach is effectively common practice.
- **Condition on Everything**: This baseline maintains only one set of posterior statistics and continuously updates them with all past data, $\boldsymbol{\eta}_t = f(\boldsymbol{x}_{1:t}, \boldsymbol{y}_{1:t})$. For recurrent network based meta-learning algorithms like the LSTM meta-learner, it is possible that the LSTM can learn to detect a task switch and reset automatically. Thus, we use this baseline only in experiments with the LSTM meta-learner to highlight how MOCA's principled Bayesian

| Class | Description | Train/Val/Test | Synsets |
|---|---|---|---|
| 1 | Non-dog animals | Train | n01532829, n01558993, n01704323, n01749939, n01770081, n01843383, n01910747, n02074367, n02165456, n02457408, n02606052, n04275548 |
| | | Validation | n01855672, n02138441, n02174001 |
| | | Test | n01930112, n01981276, n02129165, n02219486, n02443484 |
| 2 | Dogs, foxes, wolves | Train | n02089867, n02091831, n02101006, n02105505, n02108089, n02108551, n02108915, n02111277, n02113712, n02120079 |
| | | Validation | n02091244, n02114548 |
| | | Test | n02099601, n02110063, n02110341, n02116738 |
| 3 | Vehicles, musical instruments, nature/outdoors | Train | n02687172, n02966193, n03017168, n03838899, n03854065, n04251144, n04389033, n04509417, n04515003, n04612504, n09246464, n13054560 |
| | | Validation | n02950826, n02981792, n03417042, n03584254, n03773504, n09256479 |
| | | Test | n03272010, n04146614 |
| 4 | Food, kitchen equipment, clothing | Train | n02747177, n02795169, n02823428, n03047690, n03062245, n03207743, n03337140, n03400231, n03476684, n03527444, n03676483, n04596742, n07584110, n07697537, n07747607, n13133613 |
| | | Validation | n03770439, n03980874 |
| | | Test | n03146219, n03775546, n04522168, n07613480 |
| 5 | Building, furniture, household items | Train | n03220513, n03347037, n03888605, n03908618, n03924679, n03998194, n04067472, n04243546, n04258138, n04296562, n04435653, n04443257, n04604644, n06794110 |
| | | Validation | n02971356, n03075370, n03535780 |
| | | Test | n02871525, n03127925, n03544143, n04149813, n04418357 |

Table 1: Our super-class groupings for miniImageNet experiments.

runlength estimation serves to add a useful inductive bias in settings with switching tasks, and leads to improved performance even in models that may theoretically learn the same behavior.

- **Oracle**: In this baseline, the same ALPaCA and PCOC models were used as in MOCA, but with exact knowledge of the task switch times. Note that within a regret setting, one typically compares to the best achievable performance. The oracle actually outperforms the best achieveable performance in this problem setting, as it takes at least one data point (and the associated prediction, on which loss is incurred) to become aware of the task variation.
- **Sliding Window**: The sliding window approach is commonly used within problems that exhibit time variation, both within meta-learning [31] and continual learning [19, 13]. In this approach, the last $n$ data points are used for conditioning, under the expectation that the most recent data is the most predictive of the observations in the near future. Typically, some form of validation is used to choose the window length, $n$. As MOCA is performing a form of adaptive windowing, it should ideally outperform any fixed window length. We compare to three window lengths ($n = 5, 10, 50$), each of which are well-suited to part of the range of hazard rates that we consider.

## C.3 Training Details

The training details are described below for each problem. For all problems, we used the Adam [25] optimizer.

**Sinusoid**. A standard feedforward network consisting of two hidden layers of 128 units was used with ReLU nonlinearities. These layers were followed by a 32 units layer and another tanh nonlinearity. Finally, the output layer (for which we learn a prior) was of size $32 \times 1$. The same architecture was used for all baselines. This is the same architecture for sinusoid regression as was used in [16] (with the exception of using ReLU nonlinearities instead of all tanh nonlinearities). The following parameters were used for training:

- Learning rate: 0.02

- Batch size: 50
- Batch length: 100
- Train iterations: 7500

Batch length here corresponds to the number of timesteps in each training batch. Note that longer batch lengths are necessary to achieve good performance on low hazard rates, as short batch lengths artificially increase the hazard rate as a result of the assumption that each batch begins with a new task. The learning rate was decayed every 1000 training iterations.

We allowed the noise variance to be learned by the model. This, counter-intuitively, resulted in a substantial performance improvement over a fixed (accurate) noise variance. This is due to a curriculum effect, where the model early one increases the noise variance and learns roughly accurate features, followed by slowly decreasing the noise variance to the correct value.

**Wheel Bandit**. For all models, a feedforward network consisting of four hidden layers with ReLU nonlinearities was used. Each of these layers had 64 units, and the output dimension of the network was 100. There was no activation used on the last layer of the network. The actions were encoded as one-hot and passed in with the two dimensional state as the input to the network (seven dimensional input in total). The following parameters were used for training:

- Learning rate: 0.005
- Batch size: 15
- Batch length: 100
- Train iterations: 2000

and the learning rate was decayed every 500 training iterations. We allow the noise variance to be learned by the model.

We use the same amount of training data as was used in [15]: $64 \times 562$ samples. In [15], this was 64 different bandits, each with 562 data points. We use the same amount of data, but generated as one continuous stream with the bandit switching according to the hazard rate. We use a validation set of size $16 \times 562$, also generated as one trajectory, but did not use any form of early termination based on the validation set. In [37, 15] data was collected by random action sampling. To generate a dataset that matches the test conditions slightly better, we instead sample a random action with probability 0.5, and otherwise sample the action correspond to the quadrant in which the state was sampled. This results in more training data in which high rewards are achieved. This primarily resulted in smoother training.

The combined MOCA and ALPaCA models provide a posterior belief over the reward. This posterior must be mapped to an action selection at each time that sufficiently trades off greedy exploitation (maximizing reward) and exploration (information gathering actions). A common and effective heuristic in the bandit literature is Thompson sampling, in which a reward function is sampled from the posterior distribution at each time, and this sampled function is optimized over actions. This approach was applied in the changing bandit setting by [30]. Other common approaches to action selection typically rely on some form of *optimism*, in which the agent aims to explore possible reward functions that may perform better than the expectation of the posterior. These methods typically use concentration inequalities to derive an upper bound on the reward function. These methods have been applied in switching bandits in [14] and others.

We follow [30] and use Thompson sampling the main experimental results, primarily due to its simplicity (and thus ease of reproduction, for the sake of comparison). However, because the switching rate between reward functions is relatively high, it is likely that optimistic methods (which typically have a short-term bias) would outperform Thompson sampling. As the action sampling is not a core contribution of the paper, we use Thompson sampling for simplicity. Moreover, this approach meshes well with the Gaussian mixture posterior predictive (which is easily sampled from). For completeness, we present experiments in section D in which we investigate optimistic action selection methods.

**NBA Player Movement**. For this experiment, we used the LSTM meta-learner, with the encoder $\phi(\boldsymbol{x}, \boldsymbol{w})$ defined as a 3 hidden layer feedforward network with a hidden layer size of 128, and a feature dimension $n_\phi = 32$. The LSTM had a dimension of 64, and used a single hidden layer feedforward network as the decoder. ALPaCA did not perform as well as the LSTM model here; we hypothesize that this is due to the LSTM model being able to account for unobserved state variables that change with time, in contrast to ALPaCA, which assumes all unobserved state variables are task parameters and hence static for the duration of a task.

Figure 6: The performance of MOCA with ALPaCA on the sinusoid regression problem. **Bottom:** The belief over run length versus time. The intensity of each point in the plot corresponds to the belief in run length at the associated time. The red lines show the true changepoints. **Top:** Visualizations of the posterior predictive density at the times marked by blue dotted lines in the bottom figure. The red line denotes the current function (task), and red points denote data from the current task. Green points denote data from previous tasks, where more faint points are older. **a)** A visualization of the posterior at an arbitrary time. **b)** The posterior for a case in which MOCA did not successfully detect the changepoint. In this case, it is because the pre- and post-change tasks (corresponding to figure a and b) are very similar. **c)** An instance of a multimodal posterior. **d)** The changepoint is initially missed due to the data generated from the new task having high likelihood under the previous posterior. **e)** After an unlikely data point, the model increases its uncertainty as the changepoint is detected.

The following parameters were used for training:

- Learning rate: 0.01
- Batch size: 25
- Batch length: 150
- Train iterations: 5000

The learning rate was decayed every 1000 training iterations.

**Rainbow MNIST**. In our experiments, we used the same architecture as was used as in [40, 44]. It is often unclear in recent work on few-shot learning whether performance improvements are due to improvements in the meta-learning scheme or the network architecture used (although these things are not easily disentangled). As such, the architecture we use in this experiment provides fair comparison to previous few-shot learning work. This architecture consists of four blocks of 64 $3 \times 3$ convolution filters, followed by a batchnorm, ReLU nonlinearity and $2 \times 2$ max pool. On the last conv black, we removed the batchnorm and the nonlinearity. For the $28 \times 28$ Rainbow MNIST dataset, this encoder leads to a 64 dimensional embedding space. For the "train on everything" baseline, we used the same architecture followed by a fully connected layer and a softmax. This architecture is standard for image classification and has a comparable number of parameters to our model.

We used a diagonal covariance factorization within PCOC, substantially reducing the number of terms in the covariance matrix for each class and improving the performance of the model (due to the necessary inversion of the posterior predictive covariance). We learned a prior mean and variance for each class, as well as a noise covariance for each class (again, diagonal). We also fixed the Dirichlet priors to be large, effectively imbuing the model with the knowledge that the classes were balanced. The following parameters were used for training:

Figure 7: **Left**: A visualization of samples from the reward function for randomly sampled states and action $a_1$. **Middle**: The mean of the reward function posterior predictive distribution at time $t = 135$ in an evaluation run (hazard 0.02). **Right**: The run length belief for the same evaluation run. Red lines denote the true changepoints.

- Learning rate: 0.02
- Batch size: 10
- Batch length: 100
- Train iterations: 5000

The learning rate was decayed every 1500 training iterations.

**miniImageNet**. Finally, for miniImageNet, we used six convolution blocks, each as previously described. This resulted in a 64 dimensional embedding space. We initially attempted to use the same four-conv backbone as for Rainbow MNIST, but the resulting 1600 dimensional embedding space had unreasonable memory requirements for batches lengths of 100. Again, for the "train on everything" baseline, we used the same architectures with one fully connected layer followed by a softmax. The following parameters were used for training:

- Learning rate: 0.002
- Batch size: 10
- Batch length: 100
- Train iterations: 3000

The learning rate was decayed every 1000 training iterations. We used the validation set to monitor performance, and as in [5], we used the highest validation accuracy iteration for test. We also performed data augmentation as in [5] by adding random reflections and color jitter to the training data.

### C.4  Test details.

For sinusoid, rainbow MNIST, and miniImageNet, a test horizon of 400 was used. Again, the longest possible test horizon was used to avoid artificial distortion of the test hazard rate. For these problems, a batch of 200 evaluations was performed. For the bandit, we evaluated on 10 trials of length 1000. For the NBA dataset, we obtained quantitative results by evaluated on 200 sequences of horizon 150. We chose a sequence of length 200 for qualitative visualization.

## D   Further Experimental Results

In this section we present a collection of experimental results investigating task and computational performance of MOCA, as well as hyperparameters of the algorithm and modified problem settings.

### D.1  Visualizing MOCA Posteriors

Posteriors for the sinusoid and the bandit problem are provided in Fig. 6 and Fig. 7. These are visualized as they represents two ends of the spectrum; identifying changes in the sinusoid model is extremely easy, as a large amount of information is provided on possible changes for every datapoint. On the other hand, as discussed previously, only a small subset of points in the bandit problem are informative about the possible occurance of a changepoint. Accordingly, the run length belief in Fig. 6 is nearly exactly correct are concentrated on a particular run length. In contrast to this, the run length belief in Fig. 7 is less concentrated. Indeed, highly multimodal beliefs can be seen as well as the model placing a non-trivial amount of weight on many hypotheses. Finally, while some

Figure 8: Regret compared to optimal action selection for optimistic action selection with three samples (**left**) and five (**right**) samples.

Figure 9: Performance change from augmenting a model trained with MOCA with task supervision at test time (violet) and from using changepoint estimation at test time for a model trained with task-supervision (teal), for sinusoid (**left**), Rainbow MNIST (**middle**), and miniImageNet (**right**).

changepoints are detected near immediately in the bandit problem, some take a handful of timesteps passing before the changepoint is detected. Interestingly, because MOCA maintains a belief over all possible run lengths, changepoints which are initially missed may be retrospectively identified, as can partially be seen starting around time 65 in Fig. 7.

## D.2 Action Selection Schemes in the Wheel Bandit

In the body of the paper, we used Thompson sampling for action selection due to the simplicity of the method, as well as favorable performance in previous work on switching bandits [30]. However, optimism-based methods have also been effective in the switching bandit problem [14]. The MOCA posterior is a mixture of Gaussians, and thus many existing optimism-based bandit methods are not directly applicable. To investigate optimism-based action selection methods, we investigate a method in which we sample a collection of reward functions from the posterior, and choose the best action across all sampled reward models. Fig. 8 shows regret versus hazard for sampling three and five reward functions, respectively. The performance difference between MOCA and sliding window methods at low hazards is similar for Thompson sampling and for optimistic methods, as is the reversion of near-identical performance at high hazards. Compared to a standard (non-switching) bandit problem, the posterior will not concentrate to a point in the limit of infinite timesteps as there is always some weight on the prior (as the problem could switch at any timestep). This impacts optimism-based exploration methods: in the limit of a large number of samples, the prior will dominate for all states. Efficient exploration methods in the switching bandit remain an active research topic, especially paired with changepoint detection methods [30, 14, 17].

## D.3 MOCA with Differing Train/Test Task Supervision

To more closely analyze the difference between MOCA performance, which must infer task switches both at train-time and at test-time, and the oracle model, which has task segmentation information in

Figure 10: Test negative log likelihood of MOCA on the sinusoid problem with partial task segmentation. The partial segmentation during training results in negligible performance increase, while partial supervision at test time uniformly improves performance. Note that each column corresponds to one trained model, and thus the randomly varying performance across train supervision rates may be explained by simply results of minor differences in individual models.

both phases, we also compared against performance when task segmentation was provided at only one of these phases. We discuss the results of these comparisons for each of the experiments for which oracle task supervision was available below.

**Sinusoid**. Fig. 9 shows the performance of MOCA when augmented with task segmentation at test time (violet), compared to unsegmented (blue), as well as the oracle model without test segmentation (teal) compared to with test segmentation (gray). We find that as the hazard rate increases, the value of both train-time and test-time segmentation increases steadily. Because our regression version of MOCA only models the conditional density, it is not able to detect a changepoint before incurring the loss associated with an incorrect prediction. Thus, for high hazard rates with many changepoints, the benefits of test-time task segmentation are increased. Interestingly and counter-intuitively, the model trained with MOCA outperforms the model trained with task segmentation when both are given task segmentation at test time. We hypothesize that this is due to MOCA having improved training dynamics. Early in training, an oracle model may produce posteriors that are highly concentrated but incorrect, yielding very large losses that can destabilize training. In contrast, MOCA always places a non-zero weight on the prior, mitigating these effects. We find that we can match MOCA's performance by artificially augmenting to the oracle model's loss with a small weight (down to $10^{-16}$) on the prior likelihood, supporting this hypothesis.

**Rainbow MNIST**. In Fig. 9, the relative effect of the train and test segmentation is visible. Looking at the effect of train-time segmentation in isolation, comparing blue to teal and violet to gray, we see that the benefit of train-time segmentation is most pronounced at higher hazard rates. The effect of test segmentation (comparing blue to violet and teal to gray) is minimal, indicating MOCA is effectively able to detect task switches prior to making predictions.

**miniImageNet**. Fig. 9 shows that, in contrast to the Rainbow MNIST experiment, there is a large and constant (with respect to hazard rate) performance decrease moving from oracle to MOCA at test time. Interestingly, while one would expect the performance decrease with increasing hazard rate to be attributable primarily to lack of test-time segmentation, this trend is primarily a consequence of MOCA training, consistent with the Rainbow MNIST experiments. This is likely a consequence of the limited amount of data, as the trend is not apparent for the sinusoid experiment.

## D.4   MOCA with Partial Task Segmentation

Since MOCA explicitly reasons about a belief over run-lengths, it can operate anywhere in the spectrum of the task-unsegmented case as presented so far, to the fully task-segmented setting of standard meta-learning. At every time step $t$, the user can override the belief $b_t(r_t)$ to provide a degree of supervision. At known changepoints, for example, the user can override $b_t(r_t)$ to have

Figure 11: Time per iteration versus iteration number at test time. Note that the right hand side of the curve shows the expected linear complexity expected of MOCA. Note that for these experiments, no hypothesis pruning was performed, and thus at test time performance could be constant time as opposed to linear. This figure shows 95% confidence intervals for 10 trials, but the repeatability of the computation time is consistent enough that they are not visible.

all its mass on $r_t = 0$. If the task is known *not* to change at the given time, the user can set the hazard probability to 0 when updating the belief for the next timestep. If a user applies both of these overrides, it amounts to effectively sidestepping the Bayesian reasoning over changepoints and revealing this information to the meta-learning algorithm. If the user only applies the former, the user effectively indicates to the algorithm when known changepoints occur, but the algorithm is free to propagate this belief forward in time according to the update rules, and detect further changepoints that were not known to the user. Finally, the Bayesian framework allows a supervisor to provide their belief over a changepoint, which may not have probability mass entirely at $r_t = 0$. Thus, MOCA flexibly incorporates any type of task supervision available to a system designer.

Fig. 10 shows the performance of partial task segmentation at both train and test for the sinusoid problem, for the hazard rate 0.2. This problem was chosen as the results were highly repeatable and thus the trend is more readily observed. Here, we label a changepoint with some probability, which we refer to as the supervision rate. We do not provide supervision for any non-changepoint timesteps, and thus a supervision rate of 1 corresponds to labeling every changepoint but is not equivalent to the oracle. Specifically, the model may still have false positive changepoints, but is incapable of false negatives. This figure shows that the performance monotonically improves with increasing train supervision rate, but is largely invariant under varying train supervision. This performance improvement agrees with Fig. 9, which shows that for the sinusoid problem, performance is improved by full online segmentation. Indeed, these results show that training with MOCA results in models with comparable test performance to those with supervised changepoints, and thus there is little marginal value to task segmentation during training.

### D.5 Computational Performance

Fig. 11 shows the computational performance at test time on the sinusoid problem. Note that the right hand side of the curve shows a linear trend that is expected from the growing run length belief vector. However, even for 25000 iterations, the execution time is approximately 7ms for one iteration. These experiments were performed on an Nvidia Titan Xp GPU. Interestingly, on the left hand side of the curve, the time per iteration is effectively constant until the number of iterations approaches approximately 4500. Based on our code profiling, we hypothesize that this is an artifact of overhead in matrix multiplication computations done on the GPU.

### D.6 Batch Training MOCA

In practice, we sample batches of length $T$ from the full training time series, and train on these components. While this artificially increases the observed hazard rate (as a result of the initial belief over

Figure 12: Performance versus the training horizon ($T$) for the sinusoid with hazard 0.01. The lowest hazard was used to increase the effects of the short training horizon. A minor decrease in performance is visible for very small training horizons (around 20), but flattens off around 100 and above. It is expected that these diminishing marginal returns will occur for all systems and hazard rates.

run length being 0 with probability 1), it substantially reduces the computational burden of training. Because MOCA maintains a posterior for each possible run length, computational requirements grow linearly with $T$. Iterating over the whole training time series without any hypothesis pruning can be prohibitively expensive. While a variety of different pruning methods within BOCPD have been proposed [46, 38], we require a pruning method which does not break model differentiability. Note that at test-time, we no longer require differentiability and so previously developed pruning methods may be applied.

Empirically, we observe diminishing marginal returns when training on longer sequences. Fig. 12 shows the performance of MOCA for varying training sequence lengths ($T$). In all experiments presented in the body of the paper, we use $T = 100$. As discussed, small $T$ values artificially inflate the observed hazard rate, so we expect to see performance improve with larger $T$ values. Fig. 12 shows that this effect results in diminishing marginal returns, with little performance improvement beyond $T = 100$. Longer training sequences lead to increased computation per iteration (as MOCA is linear in the runlength), as well as an increased memory burden (especially during training, when the computation graph must be retained by automatic differentiation frameworks). Thus, we believe it is best to train on the shortest possible sequences, and propose $T = 1/\lambda$ (where $\lambda$ is the hazard rate) as a rough rule of thumb.

## Footnotes

[2][40] discuss this correspondence, as they outline how the choice of metric corresponds to a different assumptions on the distributions in the embedding space.

[3]The data was accessed and processed using the scripts provided here: `https://github.com/sealneaward/nba-movement-data`