[Reviews · NeurIPS 2020]

Review 1

Summary and Contributions: The authors propose a new meta-learning setting as well as a framework that can enable learning in such a setting. The setting in question is continual meta-learning using unsegmented supervised tasks. Their toy task is a continual prediction tasks where inputs and targets for a given concept are provided one after the other and the learner is then tasked to generate a model that can predict the target of the next sample correctly. Since the tasks are unsegmented there are situations where once a task shifts, the previous samples belonging to another task are no longer of as much use, and may in fact confuse the learner, therefore the authors propose a Bayesian method that can learn detect task changes such that the learner itself can incrementally become better at discerning between task changes.

Strengths: Incredible work. I think this is a very good first step towards unsegmented-task meta-learning. This is a contribution of high novelty and is strong conceptually. It opens up a whole new meta-learning setting and proposes a method that can tackle that setting. The setting itself is conceptually complex and of high usefulness, as if we want our learners to learn in environments reminiscent of the real world, the ability to learn under unsegmented tasks is key.

Weaknesses: The paper is very well done, so I have had a hard time finding a weakness. If there was one weakness I'd say that it's missing a figure that clearly contexualizes the base-model, and learner and how continual meta-learning fits in the setup.

Correctness: The claims seem correct.

Clarity: Paper is quite well written, the only problem I have is when the authors defined meta-learning as explicitly a few-shot learning method, and that would be imprecise I believe. See lines 65-67.

Relation to Prior Work: The paper makes an effort to compare to previous works, and it explicitly discusses literature related to continual learning, few-shot learning, meta-learning continual learning and continual meta-learning and how they relate to this work.

Reproducibility: Yes

Additional Feedback: This is the gist of what I got from the abstract and intro. There weren't any actual comments, but if you feel like I missed the key point of parts of the paper, then perhaps you should try to make those parts of the paper a bit more concrete. Ofcourse it also is probably my fault for having a limited attention time-slot to review this. But it might still be useful, so here you go: Abstract: Sentence 1: Intro to meta-learning. Sentence 2: Intro to task-segmented supervised meta-learning and its limitations. Sentence 3: Proposed research direction i.e. unsegmented tasks. Sentence 4: Proposed method. Detecting task change-points using Bayesian methods and doing continual meta-learning as new tasks are detected. Sentence 5: A key feature of the method. Sentence 6: Evaluation experiments applied on the method. Introduction: Paragraph 1: Intro to meta-learning for few-shot learning. Paragraph 2: Limitations of existing literature. Lack of meta-learning methods that can be trained and tested on unsegmented task streams. Paragraph 3: Research direction: To enable such methods. Paragraph 4: Contributions contains the proposed method as well as its advantages and the evaluation methods used.


Review 2

Summary and Contributions: This paper proposes MOCA, which uses BOCPD (Bayesian Online ChangePoint Detection) to infer task run lengths for a base meta-learning algorithm to use. MOCA enables meta-learning in sequences of tasks where the tasks are not explicitly segmented. Experiments show improvements over baselines on sinewave regression, wheel bandits, NBA player movement, rainbow MNIST, and miniImageNet.

Strengths: MOCA seems to be a good solution for their proposed problem setup. Through extensive experiments, the paper shows that MOCA outperforms baselines in meta-learning without task segmentation.

Weaknesses: - In all experimental settings, the paper assumes knowledge of the hazard rate lambda. We would not know lambda in a real-world problem setting in which MOCA is necessary due to missing task segmentation, so lambda would have to set manually. Experiments showing how sensitive MOCA is to this hyperparameter would have provided insight into MOCA's real-world applicability. - I'm skeptical about whether this method has many real-world applications. MOCA will only be effective in problems with (1) an unsegmented sequence of discrete tasks, where (2) task lengths follow an exponential distribution, and (3) there exists a large set of such concatenated task sequences with enough commonalities such that meta-learning is possible. The NBA player movement task seems to meet these criteria approximately, but are there more examples of such problem setups? - How necessary is the increased time complexity coming from BOCPD? Would performance degrade much if you used a simple point estimate of task length and only run meta-learning on that length?

Correctness: Please refer to overall comments

Clarity: Please refer to overall comments

Relation to Prior Work: Please refer to overall comments

Reproducibility: Yes

Additional Feedback:


Review 3

Summary and Contributions: This work extends the scope of meta-learning algorithm to the scenario where the sequential input data is not segmented into tasks. The technique used to detect task switching (referred to as MOCA) is based on a differentiable changepoint estimation algorithm in the Bayesian framework, where belief values for each possible task length are maintained and updated. The paper describes the application of the method to meta-learning algorithms that admit recursive updates (for computational savings), displaying a great performance on wide range of baselines.

Strengths: The paper addresses a novel task of extending the meta-learning to the scenario of sequential data with unknown points of task changes. Value of this task is demonstrated in a number of usecases, where, while haivng the ground truth labels available, segmentation into tasks is not trivial. The proposed solution is a sound probabilistic model based on Bayesian framework. In an nutshell, a set of meta-learning model instances, parameterized by candidate run lengths (times since last task switch) is maintained and used to compute the probabilities for each such run length. This information is used to make a soft selection over run lengths for the predicted output label, resulting in a - log likelihood expression for training the meta-learning model. So the proposed method addresses the problem in a natural way, integrating the solution within the native learning method of the meta-learner. I think the relaxation of the reqirement of data segmentation into tasks brings the few-shot learning closer to real-world applications.

Weaknesses: The performance measures for the experiments presented in Figures 3,4 is the -log-likelihood, for which the proposed MOCA algorithm has an obvious bias since its trained to minimize it. I would expect to see an additional measure for the sinusoidal regression and tracking (e.g., MSE)

Correctness: The presented probabilistic framework seems to me correct . The empirical methodology also seems adequate and convincing.

Clarity: The approach is described clearly and in detail.

Relation to Prior Work: The presented approach is clearly distinguished from the prior art.

Reproducibility: Yes

Additional Feedback: My comments were addressed in the authors response, I therefore maintain my score.


Review 4

Summary and Contributions: The authors present a meta-learning method called MOCA that handles sequence data that comprises multiple tasks but which does not require explicit information about task boundaries and instead uses Bayesian changepoint detection to estimate run lengths for each task in the sequence.

Strengths: Experiments on standard datasets suggest that the method performs well, in some cases achieving near oracle performance. The integration of Bayesian changepoint detection with meta-learning is interesting.

Weaknesses: The utility of the work is not well motivated and most of the problems are too artificial to be of real interest. It appears to be more of a solution in search of a problem than a solution to a real problem. If that's not true, then the paper should provide better motivation. The assumptions are not clearly stated, and some of the claims appear to require strong assumptions to be true.

Correctness: Appears to be good.

Clarity: I don't understand some of the details of the method or the assumptions the method requires, but overall the paper is well written.

Relation to Prior Work: The discussion of prior work is good given the limited space available.

Reproducibility: No

Additional Feedback: It would help if you could do more to motivate when data of the type MOCA is designed for would occur. It feels a little like an algorithm in search of a problem. When the task switches to a new task, does it ever switch back to a previous task, i.e., the new task is the same as one of the previous tasks? This happens at test time, but can it happen during training? If it can, can the algorithm detect that it has returned to a previously seen task? You briefly mention this in the Future Work, but I don’t understand what you’re suggesting there. Is the connection to few-shot learning necessary? Or does few-shot only mean “modest sample size” per task? Typo: extra space before the period in line 80. In line 96, is “predictive” the right word or maybe a word is missing? If you are going to use estimated density to detect task switches, isn’t it important that the data stream be randomized within tasks so that sequences of points are each iid for that task? It’s common for a dataset to be iid taken together, but changepoint detection on a series of points seems to require additional assumptions about the independence of each point in the sequence even within the same task? Given a sequence of some length L, how does the algorithm consider different numbers of tasks, e.g., all data is from one task vs. all data is from a or b or c tasks? I’m confused how the algorithm estimates the number of tasks it has seen in the sequence, i.e., the number of changepoints. Can MOCA handle the case where the input distribution does not change at all, but only the labels change. In your example about sailboats in Fig 1, if the pictures were all sailboats, but at some time the label switched from how wide is the baot to how long is the boat, would MOCA recognize the prediction task had changed even though the sequence of input images had not changed? You make a strong statement in lines 212-213 about avoiding negative transfer without giving any evidence. Avoiding negative transfer in the general case is hard, and there is probably a no free lunch theorem that says it’s essentially impossible without making further assumptions about the data generation process. What assumptions do you believe are necessary for you claim about avoiding negative transfer to be true? Just because you detect changes does not mean there can’t be negative transfer, does it? In the experimental results did you consider a train on everything approach that also has access to an input variable that specifies the position of the point in the sequence, or are you expecting an LSTM-like method to learn this automatically from the sequence?

[Author Response · NeurIPS 2020]

We thank all reviewers for their helpful comments. Our responses to each reviewer are below.

**Reviewer 1**. We thank the reviewer for the positive comments. We will add a figure explaining the full architecture in the camera-ready version of the paper.

**Reviewer 2**. The reviewer has three major critiques of the paper, which we address in order.

- **Hazard Rate**. The reviewer states that the hazard rate is assumed to be known. This is a misunderstanding. While the hazard parameter can be specified as a hyperparameter if known, it can also be chosen to be a learnable parameter of the model, since the BOCPD procedure is fully differentiable. Indeed, we use a learnable hazard rate on the NBA experiment, as there is no ground truth hazard rate. In all experiments, making the hazard parameter learnable yielded performance comparable to when it was pre-specified as a hyperparameter. We will add further discussion of this to the body of the paper.
- **Problem Setting**. As noted by reviewers 1 and 4, we believe the problem setting presented in the paper extends the domain of applicability of meta-learning tools beyond the standard setting. To showcase the broad applicability of MOCA, we included a diverse array of experiments including on real-world time series data in the NBA example, and as part of a decision making pipeline in the contextual bandit example. The reviewer's concerns center on the prevalence of settings with (1) discrete unlabeled switches in task, (2) exponential task length distributions, and (3) enough task similarity for meta-learning to be effective.
    1. Discrete switches in context are prevalent in diverse settings such as time series forecasting, mobile robotics, and anomaly detection — previous work in all of these settings has leveraged changepoint detection; MOCA leverages changepoint detection to allow meta-learning to be applied to such problems.
    2. We presented MOCA using an exponential distribution on task-length because it is simple (memoryless, single parameter) and broadly applicable. However, the changepoint detection algorithm can operate with a wide range of probability distributions for the task length [Adams & MacKay, 2007], and can be adjusted for a given application if needed.
    3. In principle, meta-learning will be useful as long as there is some commonality between the tasks, and settings in which there is absolutely no similarity between sequential tasks are rare. In robotics, for example, input data comes from the real world and is constrained by the laws of physics; meta-learning would capture these physical priors from data.
- **Point Estimate Changepoint Detection and Computational Efficiency**. MOCA relies on the full run length belief to backpropagate through the changepoint detection algorithm to train the underlying model. While "hard" changepoint detection algorithms exist, backpropagating through these would result in the standard difficulties of backpropagation through samples from a discrete distribution. At test time, one may be able to run a point estimate version of the MOCA algorithm, but the performance of this is likely highly problem-dependent.

**Reviewer 4**. We clarify that throughout the experiments, the baseline comparisons (TOE, sliding window, etc.) are trained using the same log likelihood loss function as MOCA, and so MOCA does not have a unique bias.

**Reviewer 6**. As discussed in the response to reviewer 2 ("Problem Setting"), we believe the MOCA problem setting is representative of many real world scenarios. Below, we address the other major comments from the reviewer (approximately) in order.

- **Number of Tasks and Task Recurrence**. MOCA does not directly estimate the number of tasks or changepoints in a sequence. Instead, MOCA reasons only about how much past data to use in making predictions, by maintaining a belief distribution on the run length of the current task. If the task switches to one previously seen in the sequence, MOCA would recognize that the task has changed and start adapting to the new task; MOCA would not recognize that the new task matches a previous one.
- **Label Shift**. Label changes can be handled by MOCA; this is shown for categorical outputs in our classification experiments and continuous outputs in our regression experiments. Indeed, we emphasize that MOCA can leverage changes in both the dependent and independent variable to detect changepoints.
- **IID Data Within Tasks**. We agree that this work assumes iid data generation within each task. In practice however (as demonstrated by the NBA experiment) MOCA can perform well even when this assumption is violated. Indeed, we note that many datasets have some aspect of time dependency that is ignored in the modeling. In any case, we believe our assumptions on the problem setting are stated clearly in section 2.
- **Negative Transfer**. Negative transfer is reduced because MOCA down-weights run length hypotheses that suffer negative transfer and make poor predictions. This is demonstrated via comparison to sliding window models, which do not monotonically improve with longer run lengths. We acknowledge that our statement on "avoiding" negative transfer was likely too strong; we will amend this to say that negative transfer is mitigated instead.
- **TOE with Access to Sequence Position**. The problem setting we consider is invariant under time shifts, so adding sequence position as an input to a model does not provide useful information for making predictions. Therefore, we do not consider this in our experiments.

[Meta-Review · NeurIPS 2020]

This paper addresses a continual meta-learning using unsegmented supervised tasks, which is quite a challenging and timely topic. All reviewers agree that the proposed method, referred to as MOCA, is a sound solution. The integration of Bayesian change point detection with meta-learning is an interesting idea. During the discussion period, one reviewer raised his/her score by one.